# Understanding the Reinforcement of Graphene in Poly(Ether Ether Ketone)/Carbon Fibre Laminates

**DOI:** 10.3390/polym13152440

**Published:** 2021-07-24

**Authors:** Araceli Flores, Susana Quiles-Díaz, Patricia Enrique-Jimenez, Aránzazu Martínez-Gómez, Marián A. Gómez-Fatou, Horacio J. Salavagione

**Affiliations:** 1Departamento de Física de Polímeros, Elastómeros y Aplicaciones Energéticas, Instituto de Ciencia y Tecnología de Polímeros, ICTP-CSIC, c/Juan de la Cierva 3, 28006 Madrid, Spain; quiles.susana@ictp.csic.es (S.Q.-D.); magomez@ictp.csic.es (M.A.G.-F.); horacio@ictp.csic.es (H.J.S.); 2Departamento de Física Macromolecular, Instituto de Estructura de la Materia, IEM-CSIC, c/Serrano 119, 28006 Madrid, Spain; patri_patricia91@hotmail.com; 3Departamento de Química-Física, Instituto de Ciencia y Tecnología de Polímeros, ICTP-CSIC, c/Juan de la Cierva 3, 28006 Madrid, Spain; aranmg@ictp.csic.es

**Keywords:** PEEK, graphene, carbon fibre laminates, matrix-fibre interaction, nanoindentation, electrical conductivity, structure

## Abstract

PEEK appears as an excellent candidate to substitute epoxy resins in carbon fibre laminates for high-performance aeronautical applications. The optimization of the properties and, in particular, of the transition region between the fibres and the matrix appear as a major issue prior to serial production. Graphene, modified with two compatibilizers, has been incorporated in the polymer layer with the purpose of imparting additional functionalities and enhancing the matrix-fibre interaction. It is found that both carbon fibres and modified graphene significantly influence the crystallization behaviour and smaller, and/or more imperfect crystals appear while the degree of crystallinity decreases. Despite this, nanoindentation studies show that the PEEK layer exhibits significant modulus improvements (≈30%) for 5 wt.% of graphene. Most importantly, the study of the local mechanical properties by nanoindentation mapping allows the identification of remarkably high modulus values close to the carbon fibre front. Such a relevant mechanical enhancement can be associated with the accumulation of graphene platelets at the polymer–fibre boundary, as revealed by electron microscopy studies. The results offer a feasible route for interlaminar mechanical improvement based on the higher density of graphene platelets at the fibre front that should promote interfacial interactions. Concerning electrical conductivity, a large anisotropy was found for all laminates, and values in the range ~10^−4^ S/cm were found for the through-thickness arrangement as a consequence of the good consolidation of the laminates.

## 1. Introduction

Advanced polymer-based composites have progressively substituted metals for lightweight applications in the aerospace and aeronautical industry [1,2]. In the case of high-performance structures, epoxy carbon fibre (CF) laminates have traditionally overstocked the market. The fibres bear the in-plane stresses, while the polymer matrix dominates the through-thickness properties. Epoxies are brittle and prone to cracking, and the laminates exhibit low-energy absorbance [2]. Over the past few decades, the incorporation of nanofillers to create multiscale composites appeared as a successful strategy to overcome the brittle nature of epoxies and enhance the out-of-plane properties [1,2,3,4,5]. However, epoxies demand time-consuming autoclave procedures to allow chemical reactions during moulding, and the weldability, moisture resistance, and recyclability are quite limited.

Advanced thermoplastics and, in particular, poly(ether ether ketone) (PEEK) are potential candidates to substitute epoxy resins due to the large chemical resistance and high melting temperature [6]. It has been shown that the higher toughness and ductility of PEEK compared to epoxies enhances the impact resistance of the laminates and slows down the propagation of transverse cracks [7]. At high temperatures (≈150 °C) above the glass transition temperature of PEEK, it is found that the translaminar fracture toughness substantially increases [8]. Moreover, PEEK/CF laminates exhibit better mechanical performance than the epoxy counterparts under cyclic strain at high loads [9]. However, these laminates are relatively new, large serial production is still under development, and a research effort needs to be done to fully exploit the potential of these composites in diverse research areas, including aerospace, energy storage, etc. [10,11,12,13].

One critical aspect in the development of new laminar composites is the interlaminar failure. Here, the fibre–polymer interface/interphase plays a crucial role. Preceding work using nanoindentation analysis on PEEK/glass fibre multilaminar systems showed a transition region between the matrix and the fibres, with intermediate properties that were improved with the incorporation of carbon nanotubes (CNT) and with the addition of polysulfone as a compatibilizing agent [14,15]. Enhanced interfacial properties were reported for PEEK/CF laminates by including sizing agents, such as polyetherimide (PEI) [16] or polyimide [17], together with functionalized CNTs. More recently, carbon nanotubes were also employed to strengthen the interface between the CFs and PEEK using hydroxylated PEEK-*g*-CNT as a sizing compound, and it was found that the load transfer across the laminates improved significantly [18,19].

Graphene has motivated enormous scientific interest for more than a decade as filler in polymer matrices due to its outstanding mechanical, electrical, thermal, and gas barrier properties [20,21,22,23,24]. There are many examples in the literature showing that graphene can impart significant conductivity levels, enhance thermal conductivity, and improve mechanical and barrier properties for polymer matrices of a diverse nature, expanding their applications in emerging fields, such as flexible electronics, bodily motion, energy storage, tissue engineering, etc. [20,21,22,23,24,25,26]. Compared to CNT nanocomposites, graphene nanocomposites require lower loadings for electrical percolation, and can improve thermal conductivity and mechanical performance more effectively [20,21,27,28,29].

The incorporation of graphene to PEEK has taken place at a slower pace than for other polymer matrices [30,31,32,33,34,35]. Graphene dispersion represents a challenging task due to the high melt viscosity of PEEK and the resistance to most organic solvents. Despite this, it has been demonstrated that PEEK/graphene composites can exhibit significant electrical conductivity by a careful selection of aromatic compatibilizers [30,31] or via nanocomposite powders [32]. Concerning mechanical properties, it has been shown that graphene increases modulus and hardness, and reduces creep, friction, and wear factors [27,33,35], even when commercial graphene without further modification is incorporated through a solvent-free, melt-blending process [33,35].

Studies including graphene in PEEK/CF laminates are very limited [36,37], despite the fact that the simple incorporation of pristine graphene to the PEEK layers was found to improve thermal conductivity, decrease the friction coefficient and wear rate, and enhance the flexural properties of the composites [36]. Moreover, enhanced interfacial performance was reported for PEEK/CF composites by applying graphene oxide and PEI sizing around the carbon fibres [37]. However, it is noteworthy that the selection of graphene oxide instead of graphene limits electrical conductivity and restricts the field of applications.

The present paper approaches the study of the thermal, electrical, and mechanical properties of PEEK/carbon fibre laminates with small quantities of graphene incorporated into the polymer matrix. Graphene is expected to improve the fibre–matrix interfacial interaction and provide additional functionalities to the laminate. The homogeneous dispersion of graphene in the PEEK matrix was approached in a recent paper [31], and it was shown that small quantities of PEI and sulfonated PEEK (SPEEK) compatibilizers optimized the mechanical, thermal, and electrical properties. In particular, those containing 3 wt.% of PEI and 5 wt.% of SPEEK (and equivalent quantities of graphene on each) showed the best balance of properties, approaching conductivity levels up to 10^−2^ S cm^−1^.

The objective of the present work is the study of the influence of graphene, modified with PEI and SPEEK compatibilizers, on the structure and properties of PEEK/carbon fibre laminates. The thermal behaviour, electrical conductivity, and mechanical properties are examined, and the results are correlated to structural and morphological studies. Advanced nanoindentation techniques are used to evaluate the mechanical properties of the laminates at a local scale, particularly at the boundary regions between the polymer-based layers and the fibre tows. This technique has been successfully used to probe the transition region between a thermoplastic polymer such as isotactic polypropylene and carbon fibre (CF) fabric [38].

## 2. Materials and Methods

### 2.1. Materials

A commercial powder of PEEK was used (Victrex plc, Thorton-Cleveleys, UK; PEEK 150 P, molecular weight = 40,000 g moL^−1^). Graphene was purchased from Avanzare Nanotechnology, and the specifications were 1–2 layers, with lateral dimensions of 22 ± 5 μm and 9 ± 2 μm. The plain weave CF fabric (G0904) was provided by Hexcel (Dagneux, France). The areal weight was 193 g/m^2^, and the micro-sized fibres had a diameter of ≈7 μm.

Polymer-modified graphene fillers were prepared by the dispersion of graphene in one of the two compatibilizing agents considered: polyetherimide (PEI) and sulfonated PEEK (SPEEK) [31]. The nomenclature used for the fillers was GPEI and GSPEEK, and the compatibilizer/graphene weight ratio was 55/45, as determined by thermogravimetric analysis (TGA) [31].

The nanocomposites were prepared following several mixing steps (dispersion of graphene fillers and PEEK powder in ethanol with sonication, ethanol removal, melt extrusion). Films were fabricated by hot compression, and the final thickness was around 0.35 mm. Graphene content (without the contribution of the compatibilizer) was determined to be 3 wt.% for PEEK-GPEI and 5 wt.% for PEEK-GSPEEK [31].

Details on the preparation of the polymer-modified graphene fillers and the nanocomposites can be found elsewhere [31]. Table 1 includes the melting temperature, *T*_m_, and the degree of crystallinity, *X*_c_, determined in our preceding work by differential scanning calorimetry [31].

### 2.2. Preparation of PEEK/Carbon Fibre Laminates

The laminates were prepared by alternatively placing nine plies of CF fabric and ten nanocomposite films. Consolidation of the material was made at 390 ± 5 °C in a hot press under high pressure in a two-step process, as shown in Figure 1. In the first step, five plies of CF fabric were alternated with six nanocomposite films and consolidated at the experimental conditions (temperature, pressure, time, and heating/cooling rates) indicated in Figure 1A. In the second step, two plies of CF alternated with three nanocomposite films were stacked on both sides of the laminate prepared in step 1 and consolidated using the conditions indicated in Figure 1B. This consolidation procedure was repeated twice to improve fibre impregnation. Three laminates were prepared: PEEK-GPEI/CF, PEEK-SPEEK/CF, and PEEK/CF, and their dimensions were ≈40 × 40 mm^2^ with a thickness of 1.8 ± 0.2 mm. The CF content was 70 wt.%, as determined by TGA.

Inspection of the laminates was carried out using ultrasonic C-scan, which represents a useful and non-destructive technique for detecting and quantifying defects in composite materials. ATriton 1500 (Tecnitest) equipment was employed with a 1500 × 800 mm^2^ immersion pool and different piezelectric sensors for the analysis of carbon composite materials according to Airbus Standards (Airbus AITM 6-0013). Considering that all the laminates had comparable thicknesses and carbon fibre content, changes in the observed ultrasonic signal attenuation can be correlated with differences in void contents. Figure 2 shows the ultrasonic results for the three laminates.

For nanoindentation studies, the cross section of the laminates was exposed by vertically placing a portion of the laminates (typically 10 × 5 mm^2^) using a plastic clip that was introduced in a cylindrical mould. Epoxy resin was used as the embedding medium, and the resin blocks were trimmed using a Leica microtome. The preparation of the surface included a number of polishing steps with decreasing silicon carbide paper grain size (Buehler, Lake Bluff, IL, USA) and a microcloth soaked with alumina paste (0.3 μm, Buehler, Lake Bluff, IL, USA). Figure 3 illustrates, as an example, an optical microscopy image of the PEEK-GSPEEK/CF laminate showing the alternation of CF and polymer plies.

### 2.3. Characterization 

Thermogravimetric analysis under air and nitrogen atmospheres was carried out in a TA Instruments Q50 Thermobalance (Waters Cromatografía, S.A., Cerdanyola del Vallès, Spain). The scans covered a temperature interval of 50–800 °C at 10 °C min^−1^, and the nitrogen and air flow were 60 cm^3^ min^−1^ and 90 cm^3^ min^−1^, respectively. Samples were analysed with the help of TA Instruments Universal Analysis 2000 software (version 4.5 A, Build 4.5.0.5).

Differential scanning calorimetry measurements were performed using a Perkin-Elmer DSC7-7700 calorimeter (Perkin-Elmer España S.L., Madrid, Spain). Melting temperatures and meting enthalpies were calibrated against indium (*T_m_* = 156.6 °C, Δ*H_m_* = 28.45 kJ·kg^−1^) and zinc (*T*_m_ = 419.47 °C, Δ*H*_m_ = 108.37 kJ·kg^−1^). Samples of ≈10 mg weight were extracted from the central part of the laminates and introduced in aluminium pans. Scans were carried out under an inert nitrogen flow at a rate of 10 °C min^−1^ over a temperature interval of 50 °C to 380 °C. *T*_m_ was determined as the maximum of the melting endotherm, and *X*_c_ was obtained by dividing the crystallization enthalpy of the nanocomposites (corrected for the amount of PEEK) by the value for 100% crystalline PEEK, taken to be 130 J·g^−1^ [39].

Scanning electron microscopy (SEM) images were obtained using a Hitachi SU8000 field emission microscope (Tokyo, Japan).

Mechanical properties at different locations across the multilaminar material were evaluated using a G200 nanoindenter with a low load resolution head (KLA Tencor, Milpitas, CA, USA). During the loading cycle, a constant indentation strain rate of 0.05 s^−1^ was applied, and a small oscillating force of 75 Hz was superimposed to the quasi-static loading. Based on the amplitude and phase of the displacement response to the sinusoidal loading, the contact stiffness *S* was measured continuously during the loading. The procedure involves the assumption of a simple harmonic oscillator model to describe the instrument–sample contact behaviour [40]. Finally, elastic/viscoelastic correspondence allows the relation of the stiffness to the storage modulus *E′*, following [40,41]:(1)$$\frac{E^{\prime }}{1-\nu^{2}}=\frac{\sqrt{\pi }}{2}\frac{1}{\beta \sqrt{A_{c}}}S$$

Poisson’s ratio *ν* was assumed to be 0.4 in all cases, and *β* is a correction factor that can be taken as 1.034 for a Berkovich indenter. The function describing the indenter–sample contact area *A_c_* as a function of the contact penetration depth was determined using a fused silica standard [41].

The in-plane electrical conductivity was measured using three different laminate pieces for each material (≈6 mm wide and 12 mm long), whereas the out-of-plane electrical conductivity was measured in at least eight points. The samples were previously dried under a vacuum, and silver paint was used to optimize the polymer–lead contacts. For the transverse arrangement, a terminal was placed at the top and bottom surfaces, and the resistance was measured using a digital voltmeter. For the in-plane measurements, a four-probe setup was employed, and the resistance *V*/*I* was measured with a DC low-current source (LCS-02) and a digital micro-voltmeter (DMV-001, Scientific Equipment & Services). The conductivity *σ* was determined as the inverse of the resistivity *ρ* using the equation: (2)$$\sigma =1/\rho =1/\left[4.5324\;t\;\left(\frac{V}{I}\right)f_{1}f_{2}\right]$$
where *t* is the thickness of the sample, *f*_1_ is the finite thickness correction for an insulating bottom boundary, and *f*_2_ is the finite width correction [42].

## 3. Results and Discussion

### 3.1. Influence of Graphene on the Thermal Behaviour

Figure 4 shows the thermal stability of the laminates under oxidative and inert atmospheres, as determined by TGA. Table 1 includes the characteristic degradation temperatures, together with those of the free-standing nanocomposites films taken from reference [31]. The thermogravimetric curves allowed the determination of the fibre content on each laminate, and values are indicated in Table 1. Figure 4A reveals that all laminates presented a single decomposition step under a nitrogen atmosphere that was in agreement with that found for neat PEEK, involving decarboxylation, decarbonylation, and dehydratation processes [43]. On the other hand, the degradation of the laminates under air atmosphere took place in two steps (Figure 4B). Similarly to neat PEEK, the first was related to the scission of the polymeric chains, and the second was attributed to the oxidation of the carbonaceous char formed in the first stage [44]. This second step included a weight loss of 14% due to the degradation of the CFs.

Table 1 shows that the stability of PEEK was affected by the incorporation of modified graphene (PEEK-GPEI and PEEK-GSPEEK), and the degradation temperatures at 5% weight loss *T_i_* significantly dropped in the nanocomposites. As already discussed in our preceding work, this was a consequence of the lower thermal stability of the compatibilizers [31]. However, such an effect appeared to be counterbalanced by the introduction of CFs. Indeed, *T_i_* values of all laminates were higher than those of the corresponding nanocomposites, regardless of the atmosphere. For example, *T_i_* increased by 10 °C in nitrogen and 20 °C in air for the PEEK-GSPEEK/CF laminate compared with the nanocomposite. This can be explained by the higher heat absorption capacity of the CFs with respect to PEEK. Consequently, a higher temperature is required to achieve the threshold energy needed to initiate the degradation process [45]. 

Figure 5 shows the melting and crystallization behaviour of the PEEK laminates examined by DSC. A broadening of the melting endotherms and crystallization exotherms is clearly observed when compared with neat PEEK, suggesting that smaller and/or less perfect crystals are formed in the laminates. The effect was more relevant for the laminates with modified graphene in such a way that the one with the highest graphene content showed an apparent shoulder at low temperatures in the first heating and cooling scans. Table 1 includes the degree of crystallinity and the melting temperatures obtained from the analysis of the first heating scan. Values for the free-standing nanocomposite films were also included for comparison and show that the crystallinity remained constant upon graphene addition [31]. It seemed that, in this case, the confinement effect imposed by the graphene network was counterbalanced by a nucleating effect. In contrast, lower crystallinity values were found for the laminates with modified graphene with respect to the one including neat PEEK, in agreement with preceding findings on glass–fibre reinforced PEEK/CNT composites [15]. Restrictions to polymer chain diffusion and crystal growth arising from the confinement effect of the nanofiller and the increase in the melt viscosity appeared to prevail.

Regarding the crystallization behaviour, the crystallization temperature of all laminates decreases with respect to PEEK, indicating a reduction in the crystallization rate (Figure 5B). This behaviour was already observed in PEEK/graphene nanocomposites [31], and was attributed to the restriction of chain mobility due to the graphene network and was also a consequence of the interactions between the PEEK matrix and the amorphous compatibilizers (GPEI or GSPEEK) [31].

### 3.2. Mechanical Properties by Nanoindentation

#### 3.2.1. Mapping at the Tow Front

Nanoindentation was used to evaluate the local mechanical properties of the laminates. Figure 6 (left) shows an overview of the areas selected for nanoindentation studies that covered one polymer layer sandwiched between two carbon fibre plies. The regions for mechanical mapping are marked by rectangles and include locations next to the fibre front, as well as in the middle of the polymer layer. Grids of 25 × 12 indents separated by 1 μm were used to map the mechanical properties, and the middle column of Figure 6 illustrates the *E′* values (represented by the colour code shown at the bottom of the figure) at an indenter displacement of *h* = 100 nm. Each location has an associated *E′* versus *h* plot, and Figure 7 illustrates examples of the *E′* behaviour observed. Figure 7 (left) shows *E′* vs. *h* for three representative locations in the PEEK/CF laminate (indicated by numbers): (1) at the fibre, showing a constant *E′* value with indentation depth (*E′* ≈ 55 GPa); (2) on the PEEK matrix, also displaying steady *E′* values, but significantly lower (*E′* ≈ 5 GPa); and (3) on the PEEK matrix at the beginning of the test, but coming into contact with the fibre edge during the loading cycle hence, producing a sudden *E′* rise. Figure 7 (right) illustrates the case for the PEEK-GSPEEK/CF laminate, and one can also distinguish locations at which *E′* values are independent of the indentation depth (see for example position number 4 in the fibre and numbers 6 to 8 in the reinforced matrix) and others in which *E′* increases rapidly as a consequence of the indenter–fibre contact (see location number 5).

The first observation of Figure 6 (middle) is that the *E′* maps resemble the surface topography observed by optical microscopy, and one can clearly distinguish the carbon fibres’ contour as a consequence of their higher modulus compared to PEEK. The second clear observation is that *E′* of the polymer layer increases first with the incorporation of 3 wt.% of GPEI and then, more significantly, with the addition of 5 wt.% of GSPEEK. Indeed, it is noteworthy that the *E′* map turns from light green colours into dark green ones as the quantity of graphene increases (from top to bottom). Finally, one can also appreciate that the darkest green zones tend to concentrate close to the fibre, as can be clearly discerned for the PEEK-GSPEEK/CF laminate. Let us recall that only *E′* values immediately next to the fibre edge were affected by the fibre–indenter contact during the test (e.g., location 5 in Figure 7), and the rest of the *E′* values close to the fibre (e.g., positions 6–8 in Figure 7) should be considered genuine data arising from the indentation response of the PEEK nanocomposite at that particular location.

Equivalent portions on the mechanical maps of Figure 6 (marked by black rectangles, middle column) were used for counting analysis, and the results are indicated as close to CF on the righthand side of Figure 6 (solid bars). Data associated to the fibre or to an event touching the fibre edge were not taken into account for the counting analysis. The bar charts also include the distribution of *E′* data associated to locations carefully selected to be at least 20 μm away from the fibres front, and are denoted as away from CF (dashed bars). Inspection of the results away from CF clearly show that the distribution of *E′* data shifts to higher values and widens with graphene addition. The results revealed that graphene reinforced the PEEK matrix and suggested that local variations of the quantity and orientation of graphene layers produce a broadening of the mechanical properties at the sub-micrometre scale. 

Most interesting was the comparison of the counting analysis close and far away from the fibres. For the PEEK/CF laminate, quite similar results were found, and the occurrence of an interphase between the fibre and the polymer layer cannot be distinguished at this scale. For the PEEK-GPEI/CF case, the *E′* distribution close to the fibre seemed to slightly shift to higher values with respect to the distribution far away from it. However, such a trend was more apparent for the PEEK-GSPEEK/CF laminate with higher graphene content. In this latter case, the *E′* distribution close to the fibre exhibited a tail that extended up to the 15–30 GPa interval. It is noteworthy that the high modulus tail was mostly associated to the dark green values of the mechanical map appearing immediately next to the fibre. In order to better understand the origin of the high modulus values found at the fibre–polymer boundary, electron microscopy images were taken at exactly the same location selected for nanoindentation mapping. 

Figure 8 shows two electron microscopy images of the PEEK-GSPEEK/CF laminate around the area used for nanoindentation studies (indicated by blue rectangles). The lower scale image (Figure 8a) shows a homogeneous dispersion of graphene in the polymer layer. A closer inspection of the area (Figure 8b) shows that graphene tended to accumulate at the fibre front (indicated by yellow arrows) following the fibre tow contour. It seems that the CF front acted as a filter to graphene and, in fact, one can clearly distinguish some areas inside the tows in which only the polymer layer can be seen; there is no evidence of graphene (marked by a red arrow).

The accumulation of graphene at the fibre–polymer boundary described by electron microscopy can be associated to the local mechanical enhancement detected at the tow front by indentation mapping. This higher density of graphene platelets could promote enhanced interfacial interaction with the carbon fibres (for instance, via π–π interactions) that, in the end, would appear as most beneficial for the interlaminar properties.

#### 3.2.2. Graphene Reinforcement in the Polymer Layer

The role of graphene in the mechanical reinforcement of PEEK can only be fully understood after taking into account the nanostructural changes taking place in the polymer matrix with graphene addition. Table 2 shows the average *E′* values associated to the polymer-based layers within the laminates obtained using the *E′* distribution away from CF of Figure 6. The errors represent the standard deviation over the mean values. For the sake of comparison, the average *E′* values for the free-standing nanocomposite films are also included (taken from reference [31]). Figures in brackets represent the percentage of *E′* increase with respect to PEEK, both for the free-standing nanocomposite films or as part of the laminates (i.e., 100 × (*E′_nanocomp_* − *E’_PEEK_)*/*E’_PEEK_*).

In the first place, Table 2 shows that the standard deviation associated to the average *E′* value of neat PEEK was substantially larger when the film forms part of the laminate. This finding suggests that the consolidation of PEEK between the carbon fibre plies introduced heterogeneities at the sub-micrometre scale. This result is in agreement with the broader melting endotherm observed for PEEK/CF with respect to the neat PEEK (see Figure 5). Table 2 also shows that the dispersion of *E′* data increases with the addition of graphene, both in the free-standing films and in the polymer layers within the laminates. Once again, DSC findings support this statement, and show a broadening of the melting endotherm with the introduction of graphene in such a way that a shoulder at low temperatures appears (see Figure 5). In addition, mechanical heterogeneities arising from the distribution of graphene also contribute to the broadening of the *E′* distribution, as suggested by the mapping analysis of the preceding section.

Concerning the average *E′* values, results for free-standing films showed a clear mechanical enhancement with the incorporation of graphene that can be mainly attributed to the filler reinforcement, as graphene does not produce significant differences in the degree of crystallinity (see Table 1). On the other hand, the nanocomposite layers in the laminates exhibited lower crystallinity values (≈32%) than PEEK (42%) (see Table 1), together with thinner and/or more imperfect crystal lamellae (see Figure 5). Hence, one can argue that the increment in the mechanical property upon graphene addition can be entirely associated to the filler reinforcement.

In summary, it was found that both CFs and graphene broaden the mechanical properties by introducing morphological and mechanical heterogeneities at the sub-micrometre scale. *E′* of the PEEK layer in the CF laminates improved by ≈30% upon incorporation of 5 wt.% of graphene and 5 wt.% of the SPEEK compatibilizer, and the increment can be entirely attributed to the filler reinforcement.

### 3.3. Electrical Conductivity 

Electrical conductivity appeared as the most valuable property to expand the number of applications of the PEEK/CF multilaminar materials in sectors with high requirements, such as the aerospace industry. In preceding sections, graphene was found to enhance the mechanical properties of the PEEK layer, particularly the boundary between the polymer and the fibres. It was suggested that the higher density of graphene platelets at the tow front could promote enhanced interfacial interactions with the carbon fibres, and hence should be beneficial for the interlaminar properties. The present section examines electrical conductivity along and across the multilaminar systems, and special attention is paid to the role of graphene on the anisotropic electrical behaviour.

Table 3 includes the in-plane and out-of-plane electrical conductivity values measured for the three laminates. Regarding the in-plane conductivity, quite significant values were obtained for all laminates. It is worth noting that the PEEK/CF laminate (without graphene) presented high electrical conductivity values on its surface, although the outer layer was a polymer insulating film. This observation agrees with preceding work on multiscale CNT, PEEK, and CF laminates [46], and can be explained as a consequence of the high CF content and the good impregnation of the CF plies with the polymer, yielding a very thin external polymer layer. Indeed, Figure 9 shows the optical micrograph of one selected area of the PEEK/CF laminate. It also illustrates that, at specific locations, the thickness of the outer layer was practically indistinguishable, and the CFs were very close to the outer surface. The fact that the in-plane conductivity values remained at high values upon graphene incorporation suggests that CF dominates this property.

Table 3 also shows that the out-of-plane conductivity values were significant, but several orders of magnitude lower than the in-plane values for all the laminates, corroborating the anisotropy of these systems, which is in agreement with previous studies [46]. The PEEK/CF sample presented measurable electrical conductivity, and this can be attributed to the good consolidation of the laminates. It was found that the addition of graphene produced quite small changes to the transverse electrical conductivity. It seemed that the agglomeration of graphene at the tow front, which was expected to enhance the interfacial polymer–fibre interactions, was not enough to enhance electrical conductivity. It could be that the segregation of graphene to the tow front limits electron transport in other locations far away from the matrix–fibre boundary. In addition, the fact that the fibres acted as filters to graphene and the fact that there was no graphene inside the fibre tows (Figure 8b) could also represent some limitations to electron transport.

## 4. Conclusions

PEEK/CF laminates were prepared using a two-step procedure in a standard hot press, and a good consolidation was found according to ultrasonic C-scans. Small quantities of graphene were incorporated to the polymer layer with the help of two compatibilizers, and the thermal, electrical, and mechanical properties of the laminates were examined.

It was found that the lower degradation stability of the nanocomposite films with respect to PEEK (attributed to the compatibilizers) is partly counterbalanced by the introduction of CFs. Indeed, the initial degradation temperatures under air drop by ≤15 °C for the nanocomposite laminates with respect to the PEEK laminate, compared to ≤25 °C between the nanocomposites and the PEEK free-standing films.

A broader melting endotherm at lower temperatures is found for PEEK sandwiched between CF plies with respect to the PEEK free-standing film. The incorporation of graphene further spreads the melting process over a wider temperature range, and it is suggested that both CFs and modified graphene expand the crystal size distribution to smaller and/or more imperfect crystals. 

Enhanced structural heterogeneity has a significant impact on the mechanical properties, and it is found that the dispersion of *E′* data is significantly higher when the polymer layers are incorporated to the multilaminar system. In addition, mechanical heterogeneities arising from the distribution of graphene throughout the polymer layers also contribute to the enhancement of data dispersion.

CFs and modified graphene distort the lamellar assembly, reducing the levels of crystallinity and giving rise to thinner and/or more imperfect crystals, and significant modulus improvements approaching ≈30% can be measured in the PEEK-GSPEEK layer of the CF laminate.

Indentation mapping allowed the investigation of the mechanical properties at a local scale and in particular at the polymer–fibre boundary. The study covers a research area of great importance still unexplored for PEEK-graphene/CF laminates, and provides the basis for a comprehensive understanding of the reinforcing mechanism at a macroscopic level. Indeed, a local mechanical enhancement at the polymer–CF boundary region is identified upon graphene addition that can be associated to the accumulation of the filler at the fibre front, as revealed by electron microscopy studies. This higher density of the platelets is expected to be the most beneficial for the interlaminar mechanical properties, and opens up a route for property improvements.

Finally, carbon fibres appear to dominate the in-plane electrical conductivity, and very high values have been attained for all laminates (35–90 S/cm). Transverse conductivity values are significant in all cases (2 × 10^−4^–3 × 10^−4^ S/cm), and the dispersion of graphene in the polymer layers does not seem to produce relevant differences, probably due to the poor presence of the filler inside the fibre tows. However, the conductivity levels (~10^−4^ S/cm) are adequate for a broad range of applications, including sensors, strain sensing materials, electromagnetic shielding, etc.

## Figures and Tables

**Figure 1 polymers-13-02440-f001:**
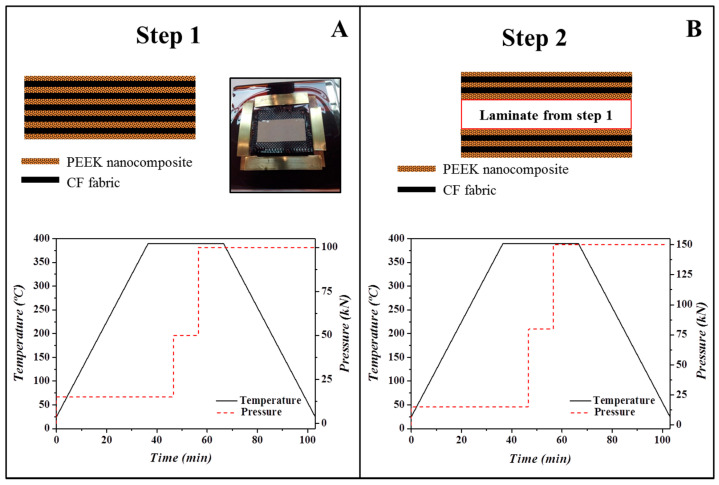
Schematic representation of the stacking sequence and consolidation procedure followed in steps 1 (**A**) and 2 (**B**) for the preparation of PEEK-based laminates.

**Figure 2 polymers-13-02440-f002:**
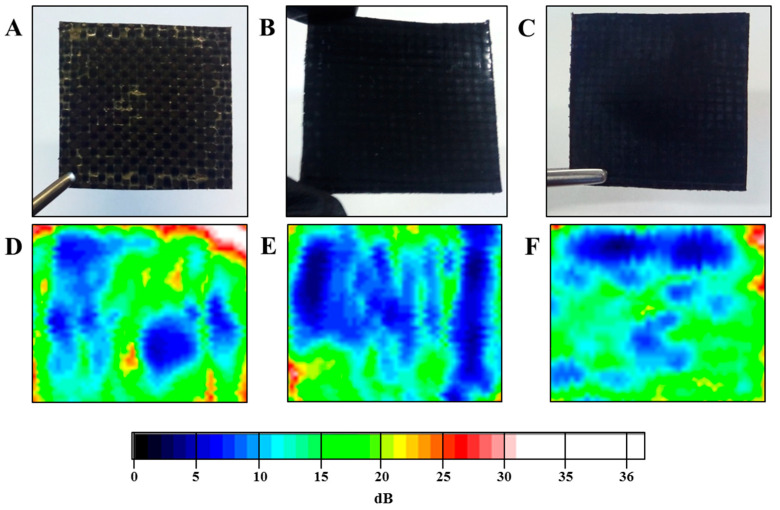
Photographs (upper row) and ultrasonic attenuation C-scans (bottom row) of PEEK/CF (**A**,**D**), PEEK-GPEI/CF (**B**,**E**), and PEEK-GSPEEK/CF (**C**,**F**). The colour scale is associated to the ultrasonic signal attenuation measured in dB.

**Figure 3 polymers-13-02440-f003:**
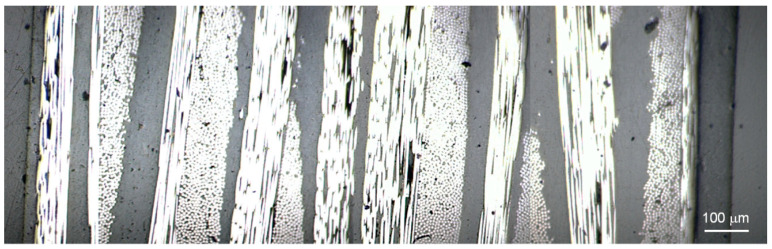
Optical microscopy image of the cross section of the PEEK-GSPEEK/CF laminate embedded in epoxy resin.

**Figure 4 polymers-13-02440-f004:**
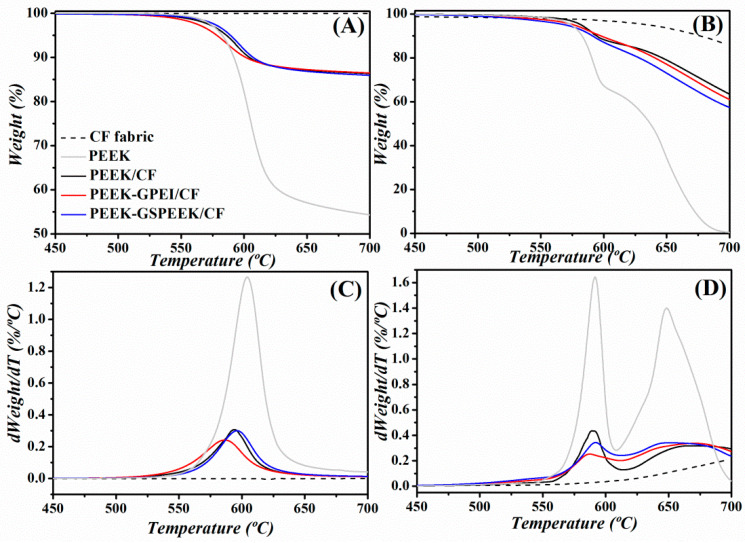
Thermogravimetric and first derivative thermogravimetric curves for the three laminates investigated, the CF fabric and neat PEEK, obtained under nitrogen (**A**,**C**) and air (**B**,**D**) atmospheres. The colour code in (**A**) applies to all data.

**Figure 5 polymers-13-02440-f005:**
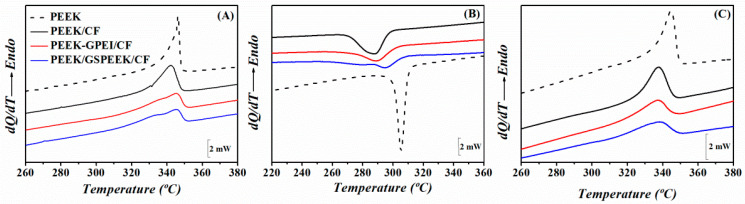
DSC scans of PEEK and the three laminates for (**A**) first heating, (**B**) first cooling, and (**C**) second heating. The colour code in (**A**) applies to all data.

**Figure 6 polymers-13-02440-f006:**
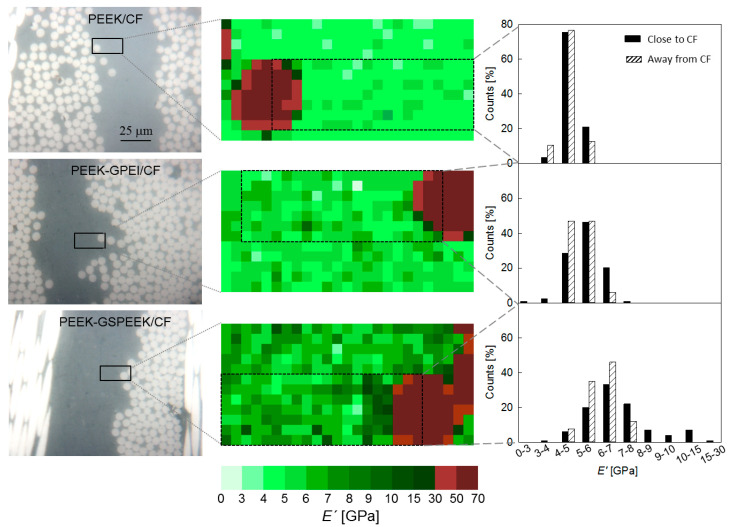
(**Left**) Microscopy images of the three laminates showing an overview of the regions selected for nanoindentation mapping (marked by rectangles). (**Middle**) Mechanical maps constructed by representing the E’ values at h = 100 nm for each location (the colour code appears at the bottom). Grids of 25 × 12 indents separated by 1 µm were used. The black rectangles denote the portion of data selected for the counting analysis close to the CF. (**Right**) Bar chart illustrating the counting analysis carried out close to (solid bars) and far away from (dashed bars) the fibres front.

**Figure 7 polymers-13-02440-f007:**
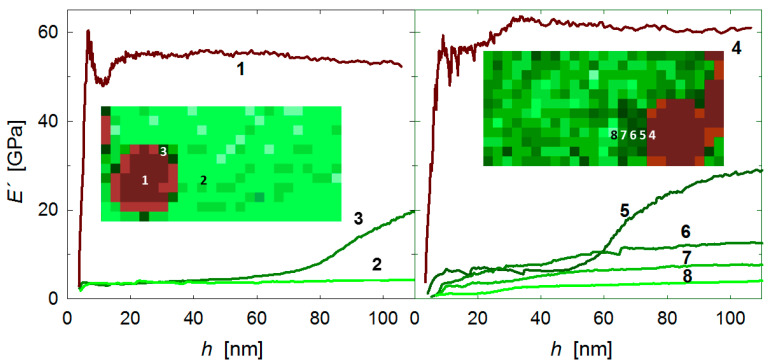
Storage modulus *E′* as a function of indenter displacement *h* for specific locations identified by numbers on the PEEK/CF (**Left**) and the PEEK-GSPEEK/CF (**Right**) laminates.

**Figure 8 polymers-13-02440-f008:**
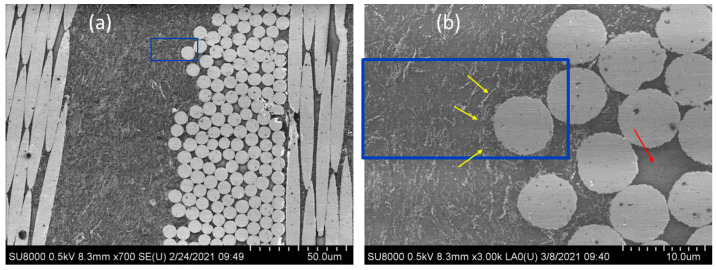
SEM images of the PEEK-GSPEEK/CF laminate showing the area selected for nanoindentation mapping (blue rectangle) at (**a**) lower and (**b**) higher magnifications. Yellow arrows indicate the accumulation of graphene at the fibre front. The red arrow denotes a region inside the tows without graphene.

**Figure 9 polymers-13-02440-f009:**
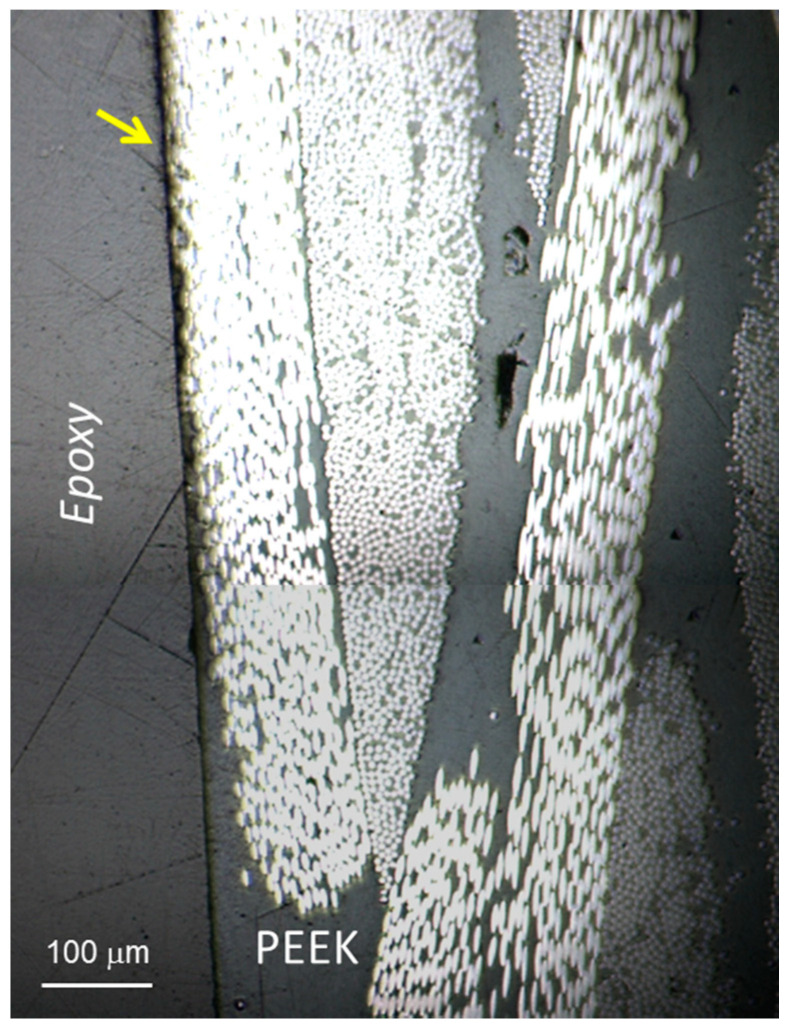
Optical microscopy image of the PEEK/CF laminate showing a very thin outer layer at specific locations (marked by a yellow arrow).

**Table 1 polymers-13-02440-t001:** Thermal parameters obtained from TGA and DSC.

Sample	G Content (wt.%) ^a^	CF Content (wt.%)	*T_i_* (°C)		First Heating	
			Nitrogen	Air	*X_c_* (%)	*T_m_* (°C)
PEEK ^b^	0		582 ± 1	574 ± 1	37 ± 2	346 ± 1
PEEK/CF	0	70	586 ± 1	582 ± 1	42 ± 2	342 ± 1
PEEK-GPEI ^b^	3		572 ± 1	566 ± 1	34 ± 2	341 ± 1
PEEK-GPEI/CF	3	67	578 ± 1	576 ± 1	32 ± 2	345 ± 1
PEEK-GSPEEK ^b^	5		579 ± 1	549 ± 1	38 ± 2	341 ± 1
PEEK-GSPEEK/CF	5	67	589 ± 1	569 ± 1	33 ± 2	345 ± 1

^a^ Graphene content in the nanocomposite film measured by TGA after heating to 800 °C. ^b^ Values are taken from reference [31]. *T_i_* = initial degradation temperature obtained at 5% weight loss. *X_c_* and *T_m_* are the degree of crystallinity and melting temperature values measured by DSC.

**Table 2 polymers-13-02440-t002:** Average indentation *E′* values for the neat PEEK and PEEK-graphene materials in the form of free-standing films (PEEK, PEEK-GPEI, and PEEK-GSPEEK) or as part of the laminates, together with CF plies (PEEK/CF, PEEK-GPEI/CF, and PEEK-GSPEEK/CF).

Sample	G Content (wt.%)	*E′* (GPa)	
		Free-Standing Film	Polymer Layer in the Laminate
PEEK	0	5.0 ± 0.1	
PEEK/CF	0		4.5 ± 0.4
PEEK-GPEI	3	5.3 ± 0.2 (6%)	
PEEK-GPEI/CF	3		5.2 ± 0.6 (16%)
PEEK-GSPEEK	5	6.4 ± 0.3 (28%)	
PEEK-GSPEEK/CF	5		6.0 ± 0.7 (33%)

**Table 3 polymers-13-02440-t003:** In-plane and out-of-plane conductivity values of all laminates. Conductivity values for the free-standing nanocomposite films are also included for comparison [31].

Sample	G Content (wt.%)	σ (S.cm^−1^)		
		Free-Standing Film	Laminates	
			In-Plane	Out-of-Plane
PEEK	0			
PEEK/CF	0		37 ± 15	(2.1 ± 0.7)·10^−4^
PEEK-GPEI	3	(2.4 ± 0.1)·10^−4^		
PEEK-GPEI/CF	3		99 ± 18	(2.1 ± 0.8)·10^−4^
PEEK-GSPEEK	5	(2.2 ± 0.1)·10^−2^		
PEEK-GSPEEK/CF	5		50 ± 16	(3.4 ± 0.8)·10^−4^

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
