# Peer review of "Understanding the Reinforcement of Graphene in Poly(Ether Ether Ketone)/Carbon Fibre Laminates"

_polymers, 2021, doi:10.3390/polym13152440_

Round 1

Reviewer 1 Report

The authors have prepared PEEK/CF laminates with small quantities of graphene incorporated to the polymer layer and studied the thermal, electrical and mechanical properties of the laminates. They found that the lower degradation stability of the nanocomposite films with respect to PEEK is partly counterbalanced by the introduction of CFs. Incorporation of graphene further spreads the melting process over a wider temperature range and it is suggested that both CFs and modified graphene expand the crystal size distribution to smaller and/or more imperfect crystals. The publication of this manuscript will provide guidance for the researchers to develop aeronautical materials. To present a high-quality publication, follow revisions are advised:

  1. The introduction should be modified. Due to its high electronic conductivity and mechanical properties, PEEK/CF has been applied as energy storage materials. In the introduction, the authors are suggested to mention this. Please refer to Energy Storage Mater. 34 (2021) 107-127, Energy Storage Mater. 27 (2020) 279-296.
  2. The language of English should be improved. There are spelling and grammar mistakes through the manuscript.
  3. The authors claimed that the dispersion of small quantities of graphene in the polymer layers is not enough to produce relevant differences in electrical conductivity. So how much graphene incorporated will lead to higher electrical conductivity?

Author Response

To present a high-quality publication, follow revisions are advised:

1. The introduction should be modified. Due to its high electronic conductivity and mechanical properties, PEEK/CF has been applied as energy storage materials. In the introduction, the authors are suggested to mention this. Please refer to Energy Storage Mater. 34 (2021) 107-127, Energy Storage Mater. 27 (2020) 279-296.

The references are now included in the manuscript (new references 12 and 13).

2. The language of English should be improved. There are spelling and grammar mistakes through the manuscript.

We have carefully checked the manuscript for misspellings and grammar mistakes according to UK English.

3. The authors claimed that the dispersion of small quantities of graphene in the polymer layers is not enough to produce relevant differences in electrical conductivity. So how much graphene incorporated will lead to higher electrical conductivity?

We are sorry for this misunderstanding. In the last paragraph of the Conclusions section, our intention was not to suggest that larger amounts of graphene should be incorporated to the polymer layer to achieve higher conductivity levels. We think that graphene quantities higher than those used in the present study (up to 5 wt.%) would easily form agglomerates. Hence, we have modified the last sentence of the Conclusions section to address that conductivity levels are high enough for a broad range of applications despite graphene does not improve the transverse conductivity possibly due to the poor incorporation inside the fibre tows.

Reviewer 2 Report

the current study investigates the effect of add graphene as a reinforcement in fibre laminates made from Poly(ether ether ketone)/Carbon Fibre. the aim here is to enhance the matrix/fibre interaction and improve the mechanical properties. The authors found that the addition of graphene can affect the crystallization behaviour and degree of crystallinity. The authors carry out mechanical tests to analyse the material and found that storage module of the PEEK increased. However, the authors also analyse the electrical characteristics and report that adding graphene does not contribute to any property enhancements.

Please consider reviewing the abstract and highlight the novelty, major findings and conclusions.

Before the last paragraph the authors are encouraged to answer the following question: What is the research gap did you find from the previous researchers in your field? Mention it properly. It will improve the strength of the article.

  1. Experimental change to “Materials and methods”

Please consider adding a list of nomenclature for all the Greek symbols and letters used in the current study at the end of the manuscript.

The article is discussed to good level and results are explained properly.

Author Response

Please consider reviewing the abstract and highlight the novelty, major findings and conclusions.

We have extensively modified the abstract according to this suggestion.

Before the last paragraph the authors are encouraged to answer the following question: What is the research gap did you find from the previous researchers in your field? Mention it properly. It will improve the strength of the article.

Following the reviewer´s recommendation, the paragraph immediately before the last one in the Conclusions section now includes the relevance of the present research in relation to preceding work.

2. Experimental change to “Materials and methods”

This heading has been changed.

Please consider adding a list of nomenclature for all the Greek symbols and letters used in the current study at the end of the manuscript.

A list of symbols and nomenclature has been added at the end of the manuscript.

The article is discussed to good level and results are explained properly.